# Human capital and regional disparities: Advancing accounting frameworks with education, health, and population dynamics

**Shuning Chen** [1]◉, **Xiangdan Piao** [2]◉, **Jun Xie** [1]‡, **Shunsuke Managi** [1]◉*

1 Urban Institute, Kyushu University, Fukuoka, Japan, 2 Faculty of Humanities and Social Science, Iwate University, Morioka, Iwate, Japan

◉ These authors contributed equally to this work.
‡ These authors also contributed equally to this work.
* managi@doc.kyushu-u.ac.jp

## Abstract

This study advances the inclusive wealth accounting of human capital (HC) to improve global research on the valuation of HC for sustainability. By innovatively integrating complex population dynamics, including schooling and labor force participation, and using a net present value (NPV) valuation method aligned with capital budgeting principles, we quantitatively measure HC in 165 countries. As a methodological advancement, we use a unified framework that incorporates education, health and economic participation via the measurement of life expectancy in different life stages to inform sustainable development investments. Our analysis from 1990 to 2020 reveals significant differences in HC development across countries. While education is strongly correlated with GDP growth, disparities in health and economic participation are critical barriers to long-term HC accumulation. Our findings argue for a comprehensive policy approach that goes beyond investing in education for its financial benefits and includes substantial improvements in health and economic opportunities to promote more equitable HC growth. We emphasize the need to incorporate complex population dynamics into HC assessments to better understand and strengthen the interdependencies between these critical factors, with the aim of reducing global development gaps.

## 1. Introduction

Human capital (HC), which is traditionally defined as the education, skills, and technology embodied in individuals, has its academic origins in the works of classical economists, such as Adam Smith. However, the pioneering contributions of Schultz [1] and Becker [2] significantly advanced the concept, framing HC as a crucial element in economic analysis. This foundation was further developed through the endogenous growth models of Romer [3] and Lucas [4] which directly linked HC to long-term economic growth. These models emphasized educational attainment and workforce skills as critical drivers of economic prosperity and received robust empirical support from studies by Lucas [4,5], Mankiw et al. [6], and Card [7].

Recent empirical research has focused on country-specific analyses of HC and its relationship with economic growth [8–10]. This body of work provides compelling evidence that

**Data availability statement:** All relevant data are included in the paper and its Supporting Information files.

**Funding:** Initials of the authors who received each award : S.M. - Grant numbers awarded to each author : JPMEERF20201001 (Environmental Restoration and Conservation Agency of Japan), JP20H00648 (JSPS) - The full name of each funder : - Environmental Restoration and Conservation Agency of Japan - Japan Society for the Promotion of Science (JSPS) - URL of each funder website : - Environmental Restoration and Conservation Agency of Japan: (https://www.erca.go.jp/erca/english/index.html) - Japan Society for the Promotion of Science (JSPS): (https://www.jsps.go.jp/english/) - Did the sponsors or funders play any role in the study design, data collection and analysis, decision to publish, or preparation of the manuscript? : The funders had no role in study design, data collection and analysis, decision to publish, or preparation of the manuscript.

**Competing interests:** The authors have declared that no competing interests exist.

improvements in educational attainment are strongly correlated with economic development [11–20]. For example, Angrist et al [12], utilizing global learning data from 164 countries between 2000 and 2017, highlighted the positive effects of HC on economic growth. Similarly, regional studies have shown that the impact of education on growth varies: in East Asia, primary and secondary education have a stronger effect, whereas tertiary and vocational education are more impactful in South Asia [15,21].

Recently, the need for a comprehensive methodology that captures both market and nonmarket values of human capital has been emphasized [12,22–24]. Current HC valuations, which focus primarily on economic outcomes, often overlook the broader contributions of HC to sustainable development. Leading global institutions, such as the World Bank and the United Nations, have emphasized the critical role of HC in achieving sustainable development, as reflected in initiatives such as the Sustainable Development Goals [25] and the World Bank's Human Capital Index [26].

Two major components must be considered when building a sustainability-integrated framework for HC accounting. First, health is a fundamental aspect of HC. Grossman [27–29] identified health as a consumer good with immediate benefits and as a long-term investment that enhances productivity. This dual role has been supported by subsequent research, which has highlighted the synergy between health and education in strengthening HC [30–32]. Nevertheless, empirical assessments that explicitly link education and health remain limited, partly owing to data challenges and the complexity of modeling these interactions.

Second, the integration of population dynamics—including age structure, mortality rates, and migration patterns—into HC models significantly advances the understanding of the economic implications of demographic changes. The research by Lee and Mason [33] and Lutz and KC [34] showed the importance of demographic factors in shaping HC investments. Moreover, studies on the intergenerational transmission of HC [35,36] and the effects of global challenges, such as migration [37], underscore the need for comprehensive modeling of population dynamics to fully capture the impact of HC on economic and social outcomes.

Despite notable progress, few studies have offered a comprehensive, global analysis that integrates these key factors. Lim et al. [38] developed a holistic measure of HC that includes education, health, and population dynamics and provided valuable insights across 195 countries from 1990 to 2016. Similarly, Campbell and Üngör [39] explored the relationship between HC and income disparities and suggested a framework that accounts for health, schooling, and cognitive skills. However, challenges remain in the valuation of HC, particularly due to variations in educational quality and data comparability across contexts [22].

Building on the conceptual framework proposed by Dasgupta [40,41] and Arrow et al. [8], we view HC as a cornerstone of sustainable human welfare. HC, consisting of a population's health and education, represents the balance between investment and returns. Following this approach, we aim to update the HC accounts presented in the Inclusive Wealth Report [42,43] by incorporating complex population dynamics into HC estimation. Specifically, for 165 economies from 1990 to 2020, we empirically assess HC, investigate its relationship with economic growth and identify the factors driving regional variations in HC.

The primary findings reveal significant geographical disparities in HC, with higher levels observed in the Global North, whereas the growth rate of HC is more pronounced in the Global South. Additionally, although HC and GDP are positively correlated, the strength of this relationship varies by region. In the G20 countries and high-income countries, the correlation is particularly strong, with education playing a key role in economic growth. In contrast, the noneducational components of HC show a smaller impact on growth, which

may explain the weaker overall influence of HC in regions lacking balanced HC development. Finally, the advancement of HC relies on the collective progress of its key components—life expectancy, education, and economic participation—indicating that improvements in a single aspect are insufficient to drive HC growth.

This paper contributes to the literature by proposing a revised lifetime income approach that flexibly identifies dynamic population stages and evaluates HC through estimations of school life expectancy (SLE) and work-life expectancy (WLE). This approach enhances our ability to assess the long-term sustainability of HC and moves beyond a narrow focus on economic benefits.

The updated HC accounts for 165 countries demonstrate how demographic changes relate to sustainable development, offering valuable insights for national and international policy planning. Specifically, the findings emphasize the need for a holistic approach to HC development, with balanced investments in education, health, and demographic factors to ensure long-term economic prosperity. Considering their unique demographic contexts, developing countries should adopt targeted policies to address gaps in education and health.

The next section delves into the theoretical foundations and methodologies underpinning the updated HC accounts. Section 3 presents empirical results that illustrate the relationship between HC accumulation and economic development across 165 countries from 1990 to 2020 and identifies long-term trends in HC accumulation. The subsequent discussion section provides a detailed analysis of these results, highlighting key insights and findings. Finally, the paper concludes with a discussion of the broader implications of these HC accounts for advancing global HC assessments, as well as their practical applications and implications for shaping social policy.

## 2. Method

### 2.1. Revised lifetime income-based approach

From the perspective of inclusive wealth, the wealth of an economy is measured by including all capital assets that support sustainable development. HC, as one of the three pillars of capital assets [8,44], is composed of two key components: the measure of quantities and the accounting price (or shadow price) of the capital asset. The accounting price reflects the true value of capital assets and is equal to the net present value of future capital services that can be obtained from the relevant assets (see Supporting Information S1 File).

Two approaches for valuing human capital stock have received widespread attention [45]. In the "cost-based approach", human capital stock is calculated as the depreciated value of past investment streams, including individual, household, employer, and government investments [1,46,47]. In the "lifetime income approach", human capital is calculated by summing the discounted present value of all future income streams that individuals in the population are expected to earn during their lifetime [48–53]. While these two approaches are, in principle, equal despite differences in statistical calculations and valuation [54], the net present value (NPV) estimation from the lifetime income method is preferable from a theoretical perspective. This is because it captures all future returns allocated to the relevant assets, replicating the market-equivalent valuation and aligning with the wealth accounting framework.

Inclusive wealth human capital accounting is based on the revised Jorgenson and Fraumeni (J–F) lifetime income approach [55]. In the original J–F approach, the population is grouped into stages by fixed age and status on the basis of detailed survey data. In our analysis, we apply a more flexible grouping of population stages:

1. First Stage: Ages 0–4, which represent the dependent population before receiving education. The human capital of this group is considered unproduced and assumed to be zero.

2. Second Stage: The population receiving public education, with school life expectancy (SLE) denoted as *s*. This group comprises individuals aged *5* to *5+s*, representing the current public investment in education, who are not yet included in the stock of human capital. On the basis of Becker's [56] equilibrium model of human capital investment and returns, the average education level of this population is measured by SLE. According to the Mincer formula, with φ representing the interest rate of education, the average level of labor force human capital is defined as ([57,58]:

$$h = e^{\varphi S} \tag{1}$$

3. Third Stage: Once individuals complete public education, they are considered part of the current stock of human capital. The total quantity of human capital is thus estimated as:

$$H = e^{\varphi S} \cdot \int_{5+s}^{\infty} N(n) dn \tag{2}$$

where $N(n)$ represents the number of people aged *n*.

Next, we use the unit accounting price for human capital, denoted as $\lambda_H(t)$, and the average remaining working years for receiving education compensation, denoted as *T*, to estimate the accounting price of human capital assets:

$$p_h = \int_0^T \lambda_H(t) e^{-\rho t} ds \tag{3}$$

where ρ is the discount rate. Different from the original J–F approach, our method does not account for human capital that has not yet been produced. Additionally, we factor in market equilibrium between education investment and lifetime returns. Moreover, the quantity of human capital includes the entire population that has completed education, not just the working-age population.

By adding the accounting price of human capital to the stock, we obtain the value of human capital assets as

$$HC = Hp_h = e^{\varphi S} \cdot \int_{5+s}^{\infty} N(n) dn \cdot \int_0^T \lambda_H(t) e^{-\rho t} ds \tag{4}$$

To evaluate sustainability and exclude the population diffusion effect on wealth, the analysis of per capita human capital $\dfrac{HC}{\int_0^{\infty} N(n) dn}$ is favored. Furthermore, we compare the contributions of education level h, the labor ratio of the population $\dfrac{\int_{5+s}^{\infty} N(n)}{\int_0^{\infty} N(n) dn}$ and prices $\int_T^0 \lambda_H(t) e^{-\rho t}$ to changes in per capita human capital.

Although the net present value method has been proven effective, assumptions about the average (remaining) lifetime of relevant assets and sufficient predictions for the valuation of the assets are crucial. Next, we explain the lifetime assumptions and valuations of human capital assets with the consideration of health and population dynamic demographics.

## 2.2. Estimation of life stages considering health and population dynamics

For the unit accounting of human capital, we consider both direct income and the indirect compensation of human capital on the basis of dynamic equilibrium, and the theoretical calculation of the accounting price of unit human capital is expressed as follows (see supporting information S1 File):

$$\lambda_H(t) = N(t) u(t) \cdot \frac{\partial F(K(t), H(t), BL(t))}{\partial H(t)}. \tag{5}$$

where u(t) represents the average welfare flow per person, which is derived from the products and services from production function F. The accounting price of human capital reveals the marginal value of human capital in production, as the complete market condition has been satisfied.

The sustainability condition of wealth accounting is that the total amount of capital assets does not decrease under constant accounting prices. Nevertheless, the assumption of long-term invariance in human capital asset accounting prices is unrealistic. Thus, the trend changes in accounting prices over time are considered [8]. In addition, Dasgupta [44] discussed the enabling assets contained in human capital, including the impacts of health, social capital, and institutional capital on the accounting price of human capital, which are related to the actual use of human capital in social and economic activities.

Therefore, the definition of the staged population in the lifetime income approach is essential to ensuring that human capital accounting correctly reflects market and nonmarket benefits. We incorporate schooling rates, employment rates, survival rates, and the actual structure of the population into the human capital accounting process. To assess the average education level, we use a Markov model [59,60] and explain the SLE as follows:

$$s = \frac{\int_5^{24} S(n) R_s(n) dn}{S(n)} \tag{6}$$

where S(n) indicates the survival function at age n and where $R_s(n)$ indicates the probability of the population aged between 5 and 24 years being enrolled in school. Thus, equation (3) indicates that participation in education and health status affect the real level of unit human capital.

We consider two statuses of individuals in the population stage following the completion of public education. One is the active participation in economic activity and in the job market, and the other is participation in only consumption. By incorporating the probability of participation in the job market, survival rate, and population structure, we can calculate the work–life expectancy (WLE) for all ages and then measure the average remaining working years for the total population expressed as

$$T = \frac{\int_{5+s}^{\infty} N(n) WLE(n)}{\int_{5+s}^{\infty} N(n) dn} \tag{7}$$

where $R_w(n)$ indicates the probability of job participation at age n and where

$\dfrac{\int\limits_{\infty}^{5+s} S(n) R_w(n) dn}{S(n)}$ represents the WLE() of the population at age n. Equation (6) estimates T,

which includes the impact of health and job status, as well as the population structure. A large, inactive population due to aging leads to an increase in the denominator and a decrease in T.

This human capital asset provides a flexible accounting framework that reduces data requirements and systematically combines health and population dynamics. Next, we explain the actual accounting process.

## 2.3. Data, adjustment, and limitations

Owing to data limitations, our estimates of SLE and WLE are based on the life table circuit method (see Supporting Information S1 File). This method was used in studies by Stockwell and Nam [61], Brookshire and Cobb [62], and Brookshire [63]. Survival probabilities, school

participation rates, and employment rates are estimated via this method, which is a special version of the Markov model [64]. The required cross-national data are relatively accessible.

The population age structure data N(n) and mortality risk rates used in our HC accounting are taken from the United Nations Population Statistics [65]. Data required for cohort enrolment probabilities are provided by UNESCO Education Statistics (UIS) [66]. They include gross enrolment rates and enrolment rates for primary, secondary, and tertiary schooling by gender. These enrollment rates are adjusted to account for each country's entry age and education system, with the starting age standardized to age 5 (see Supporting Information S1 File). Labor force participation rates are obtained from the International Labour Organization (ILO) [67].

We estimate the marginal productivity per unit of HC based on the GDP by integrating GDP data in constant USD from the United Nations (2015), the marginal return to labor from the EORA input–output matrix [68], and the total labor input from the ILO, following methods introduced by Arrow et al. [8] (2012) and used in previous Inclusive Wealth Reports [50,51,69]. The accounting price of unit human capital is estimated as the average value in the research period to exclude the impact of fluctuations over time on human capital.

In the next section, we present cross-country HC accounting results and discuss the findings.

## 3. Results

### 3.1. Disparity of HC growth

Fig 1 provides a comprehensive set of visualizations organized by octile to illustrate the global distribution of HC from 1990 to 2020. Specifically, Fig 1A shows the distribution of HC in 1990, whereas Fig 1B shows the HC per capita for the same year. Fig 1C shows the rates of change in HC in different octiles between 1990 and 2020. Consistent with other HC estimates [33,70], the results show that the highest HC values are in North America, Western Europe, East Asia and the Pacific, accounting for the top 12.5%, and significantly higher than those in Africa and Central Asia. Sub-Saharan Africa, South Asia, and Central Asia have the lowest HC. Fig 1B shows that the highest HC per capita levels are predominantly found in the Global North, such as North America, EU countries, Japan, and some oil-exporting countries in the Middle East. Fig 1C shows that the increase in HC tends to be inversely proportional to its initial level in 1990; countries with the highest initial HC levels, such as Japan, experienced the smallest changes or negative growth, while the largest increases were observed in Africa, Central Asia, and South America.

### 3.2. Relationship between GDP and HC

Fig 2 explores the relationship between GDP per capita and HC in 1990 and 2020, as well as the correlation between the changes in both variables during this period. Fig 2A shows a positive correlation between the quartile of HC and GDP per capita in 1990, indicating that countries with higher HC levels tended to have higher economic output. This relationship remained consistent in 2020, as shown in Fig 2B, although the disparity in GDP levels increased within quartile groups. The GDP per capita of countries in the highest HC quartile more than doubled from 1990 to 2020, whereas countries in the lowest quartile presented stagnant GDP per capita.

Fig 2C shows the relationships between changes in HC and GDP per capita from 1990 to 2020. Despite the positive correlation between HC and GDP levels in both 1990 and 2020, the growth in GDP did not always correspond to proportional increases in HC. This suggests a weak correlation between HC growth and GDP growth over time and indicates that HC

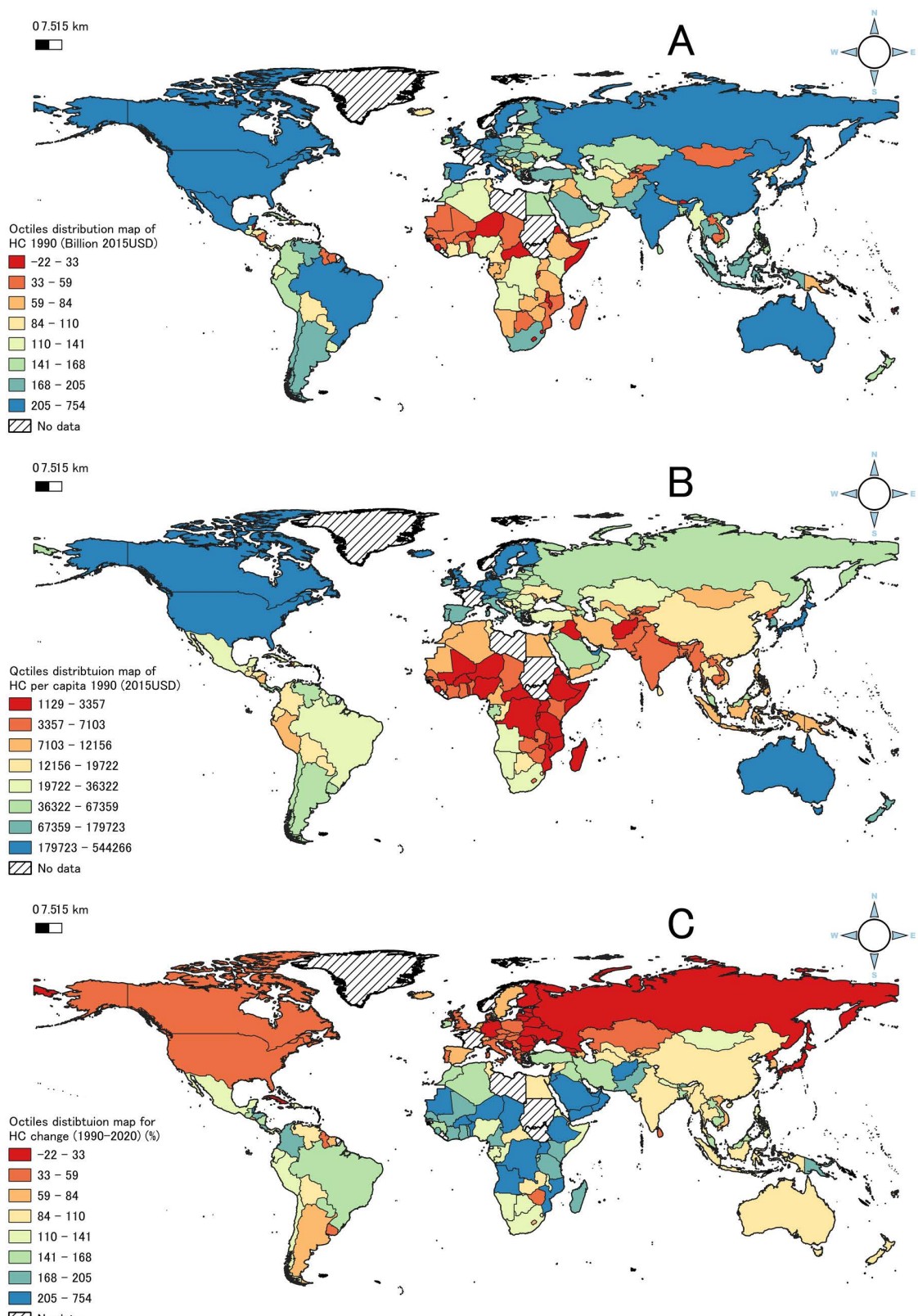

**Fig 1. Octile distributions of the global HC, global HC per capita, and HC growth from 1990 to 2020.**

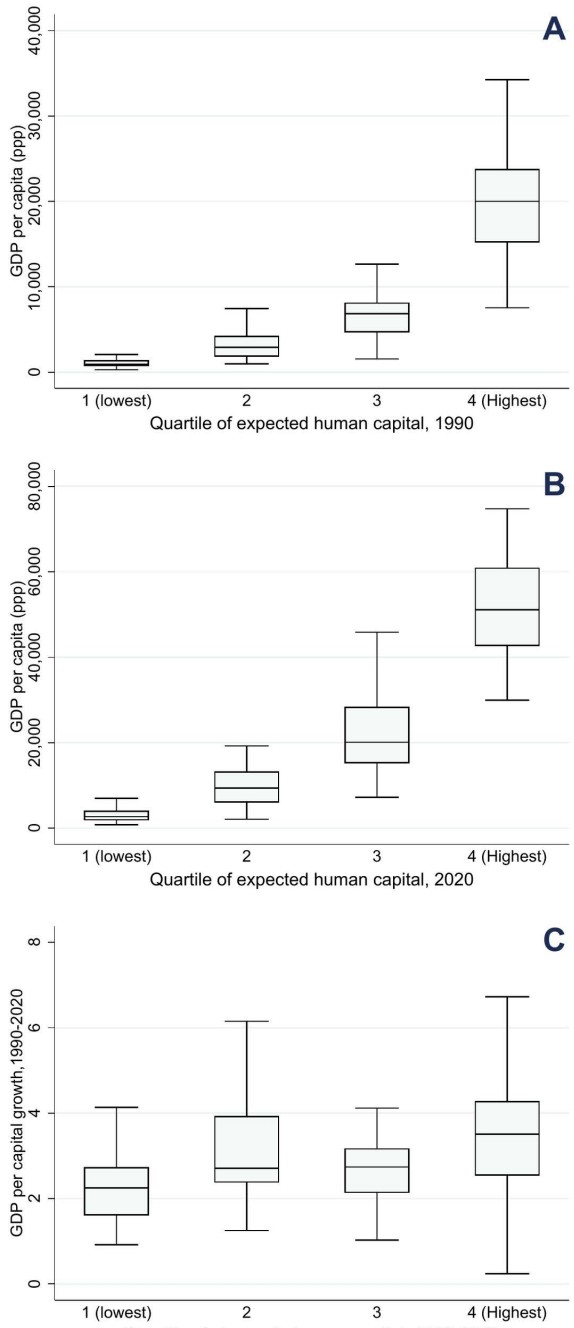

**Fig 2. Quartile of change in HC and GDP per capita from 1990 to 2020.**

accumulation may not have been efficient relative to economic growth. These results highlight that while HC and GDP levels are related, the drivers of GDP growth may not be directly linked to HC improvements, pointing to inefficiencies in how HC contributes to broader economic outcomes.

Fig 3 explores the relationship between GDP per capita and HC per capita across different geographical regions in 1990 and 2020. The World Bank's regional classification system is used in the analysis, categorizing 165 countries and including the G20 and high-income countries

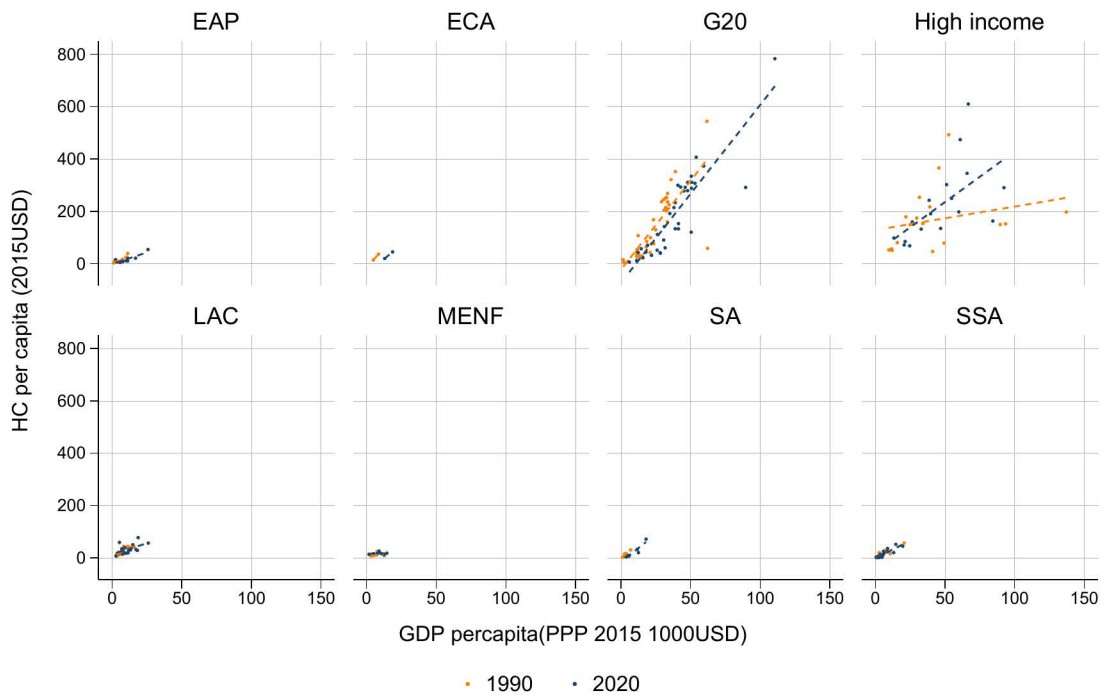

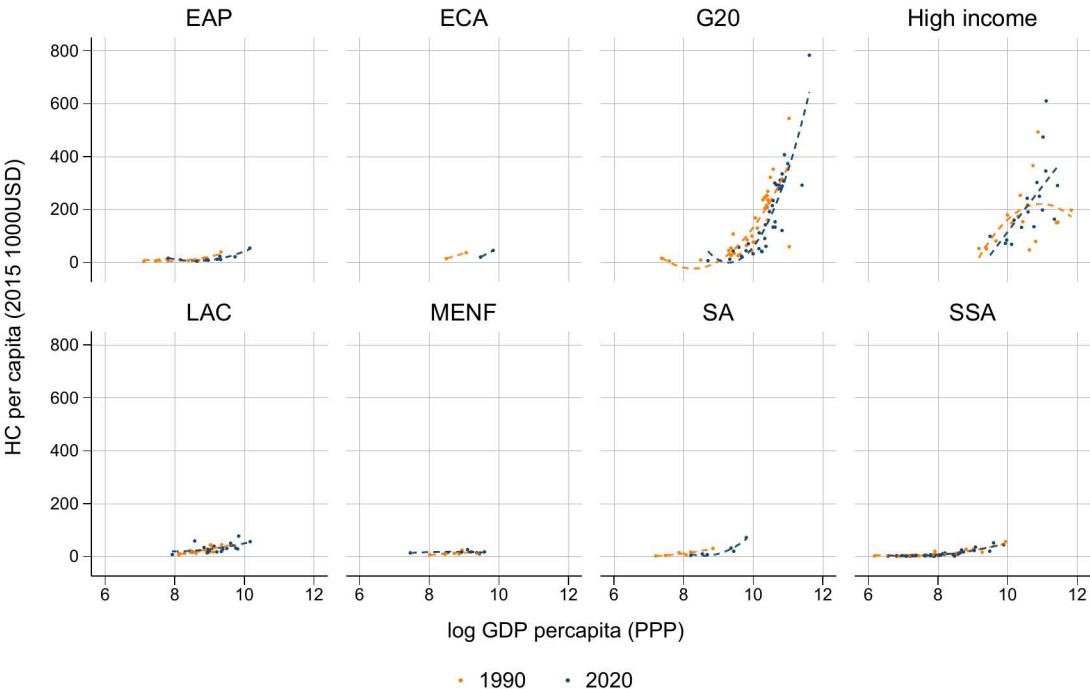

**Fig 3. Relationship between HC per capita and GDP per capita in 1990 and 2020.**

as supplementary groups. This allows a more detailed examination of growth patterns across geographical and economic regions (see Supporting Information S1 File for detailed group classifications).

Fig 3A reveals a persistent positive relationship between HC and GDP per capita in both 1990 and 2020, highlighting the role of HC in driving economic prosperity despite global economic fluctuations. However, by 2020, the correlation between HC and GDP had weakened in most regions, except in high-income countries, where a strong relationship persisted. This suggests that GDP increases are not always matched by proportional HC growth, especially outside the high-income group.

In Fig 3B, we further refine this analysis by using log-transformed GDP data to account for differences in GDP scale. While a positive correlation is observed between GDP and HC in the G20 and high-income countries, other regions—such as Latin America and the Caribbean (LAC), the Middle East and North Africa (MENA), South Asia (SA), and sub-Saharan Africa (SSA)—show disproportionately low HC accumulation, even in cases of GDP growth. This reflects ongoing challenges in human capital development, especially in regions that have economic output comparable to that of some G20 countries but lag behind in HC accumulation.

### 3.3. Components of HC and their relationships with GDP growth

Fig 4 shows how different components of HC contribute to overall HC growth and their relationships with GDP growth over three decades. Fig 4A compares overall HC growth with GDP growth, setting the stage for further component-level analysis. Fig 4B shows a close correlation between growth in school life expectancy (SLE) and GDP growth, suggesting that investments in education are significantly linked to economic performance. This finding reinforces the importance of educational attainment as a critical factor for economic

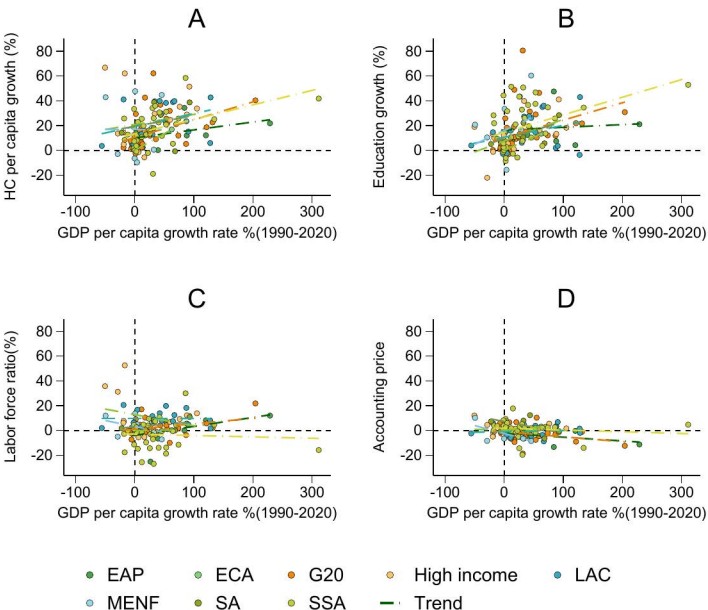

**Fig 4. The relationship between the per capita growth of HC components and GDP per capita growth from 1990 to 2020.** (a) Indicates the relationship between the growth rate of GDP per capita and HC per capita. (b) Indicates the relationship between GDP per capita growth and the contribution from education. (c) Indicates the relationship between GDP per capita growth and the contribution from educated labor. (d) Indicates the relationship between GDP per capita growth and the contribution from remaining years of work.

development. Fig 4C illustrates the contribution of the adult population proportion to HC growth, revealing an inconsistent relationship with GDP growth. In sub-Saharan Africa (SSA)—and to a lesser extent in regions such as the Middle East and North Africa (MENA), South Asia (SA), and even high-income countries—GDP growth does not translate into proportional increases in the contribution of the adult population to HC. This discrepancy underscores significant challenges in SSA, including high fertility rates and low health standards, which severely impede effective HC accumulation. In contrast, regions such as East Asia and the Pacific (EAP) and Eastern and Central Europe (ECA) demonstrate more stable responses to GDP growth, reflecting a more consistent contribution of the adult population proportion to HC growth. Fig 4D explores the impact of price, affected by the remaining years of work on HC growth, and shows a weak or negative correlation with GDP growth. This suggests that per capita GDP growth does not consistently lead to proportional gains in lifetime earnings, indicating inefficiencies or disparities in the wealth distribution. Additionally, while regions such as LAC, ECA, and high-income countries show a positive relationship between remaining work years and GDP growth, other areas do not exhibit the same correlation, indicating that GDP growth does not necessarily drive increased status in job participation.

### 3.4. The impacts of education, health, and economic participation on HC

Fig 5 explores the complex relationships among life expectancy, education, and average remaining working years across countries from 1990 to 2020, highlighting how these factors contribute to HC accumulation.

Fig 5A shows that increases in HC per capita in most countries have accompanied general improvements in life expectancy over the past 30 years. However, the magnitude of improvement varies significantly, particularly in sub-Saharan Africa (SSA), where HC growth has lagged despite health improvements.

Fig 5B shows a positive correlation between improved education and increased life expectancy, indicating that countries with better education systems also tend to experience improvements in health outcomes. This further underscores the interconnectedness of education and health in shaping HC.

Fig 5C illustrates the inconsistency between the increase in years of education and remaining working years. For example, in EAP and ECA, the remaining working years decrease as the years of education increase, revealing an inverse relationship. In contrast, in regions such as LAC, MENF, and high-income countries, years of schooling are positively related to remaining working years. However, no clear relationship between education and remaining working years is observed in SSA.

Fig 5D shows that a reduced expected working life, particularly in EAP and ECA countries, is a common issue. Although these regions experience longer life expectancies, they do not necessarily translate into more working years. In the G20, South Asia (SA), and SSA, increased life expectancy is not always associated with more working years. This suggests that factors such as lower economic performance and employment difficulties may contribute to lower economic participation, reducing the potential HC contribution from the working-age population.

Fig 6 presents the average change in HC per capita across two periods: 1990–2005 and 2005–2020. The growth factors are broken down into education level, adult population share, and remaining working years.

Period 1990–2005: During this period, education was the main driver of HC growth, particularly in the G20 and high-income countries, where significant gains were observed. In high-income countries, an increase in the adult population share further boosted HC growth,

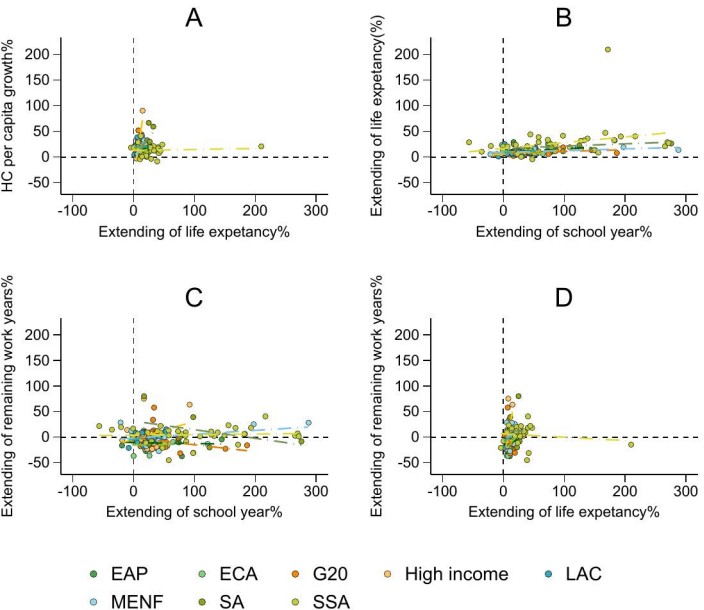

**Fig 5. Comparison of the change in HC per capita, life expectancy, school life expectancy, and remaining years of work.** (A) Indicates the relationship between the growth rate of life expectancy and school life expectancy. (B) Indicates the relationship between the growth rate of life expectancy and school life expectancy.(C) Indicates the relationship between the growth rate of school life expectancy and remaining working years. (D) Indicates the relationship between the growth rate of life expectancy and remaining working years.

although reductions in remaining working years tempered these gains. Most regions, except ECA, experienced HC growth driven by education, whereas South Asia (SA) experienced balanced growth across education, the share of adults, and remaining working years. Sub-Saharan Africa (SSA) experienced the smallest increase in HC per capita.

Period 2005–2020: In this period, the contribution of education to HC growth diminished, particularly in the G20 and high-income countries. The largest increase in HC per capita in high-income countries was driven by the increase in the share of the adult population, but the reduction in remaining working years offset this increase. Education was also a key factor in ECA countries, although cuts in working years negatively affected HC accumulation. Across most regions, the second period highlights the fact that HC accumulation driven by factors other than education is unstable, as the contribution of education declined, and other factors played a less prominent role in HC growth.

## 4. Discussion

This study updates the HC account from the recent Inclusive Wealth Report [42,43] and presents the HC accounting approach on the basis of the integration of health and population dynamics modeling. The analysis from 1990 to 2020 reveals significant differences in HC development among 165 countries. In this section, the above results are discussed under three main points.

First, we confirm the discrepant relationship between HC and economic development, which is consistent with that shown in previous studies [11–20]. However, our analysis shows that while HC has grown significantly over the past three decades, this growth has not been exactly correlated with GDP growth. In particular, there are disparities between different regional groupings, with the gap between high-income countries and economically robust

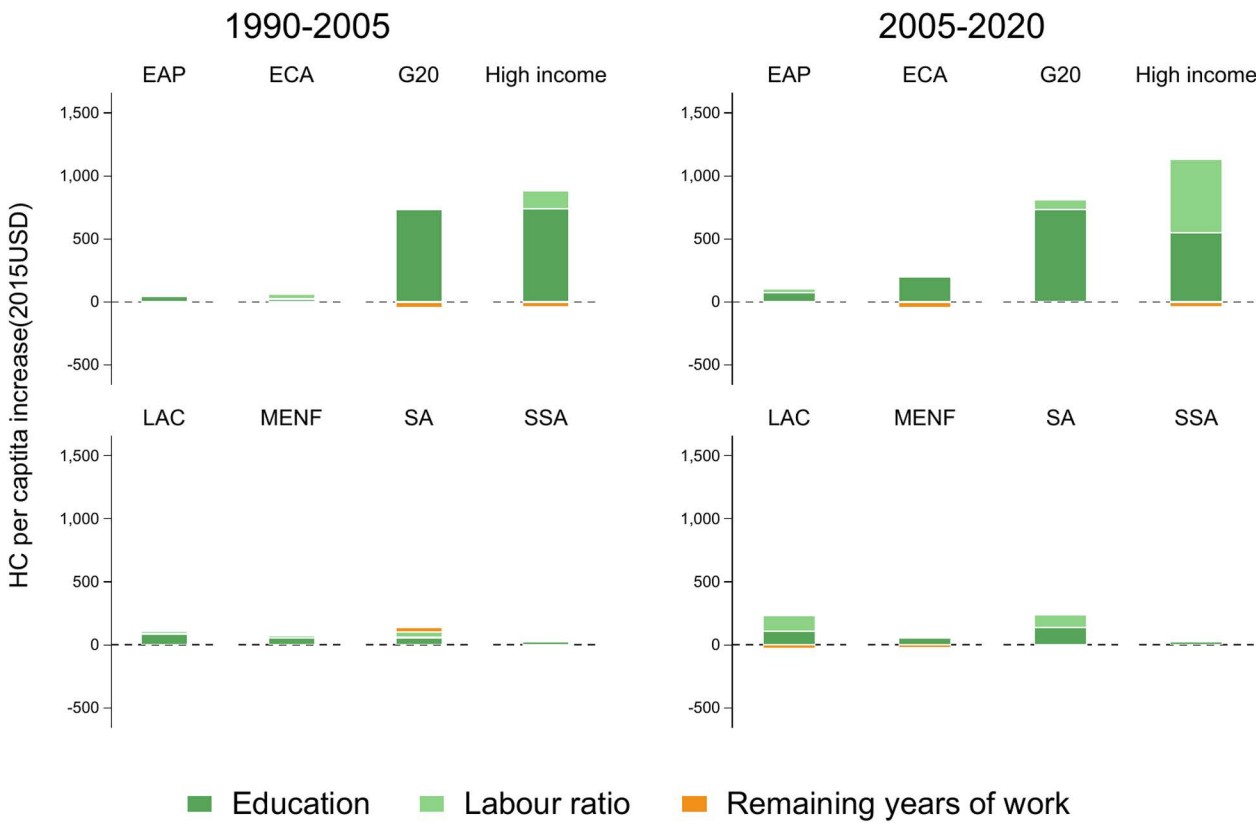

**Fig 6. Mean value of the contribution of HC components to growth in HC per capita by region.**

G20 countries widening relative to other regions. This discrepancy suggests that factors other than HC may drive GDP growth or that the impact of HC on GDP is more complex than previously understood.

Second, the impact of noneducational components on improving HC needs to be considered. While improvements in education are strongly correlated with GDP growth, the health-related components of HC and the dynamics of the economic participation of the population lag significantly behind, especially in less developed regions, such as South Asia and sub-Saharan Africa. Mortality risks, including diseases and environmental factors, such as air pollution, may be critical aspects of these disparities. This lag suggests that more than improvements in education are needed to promote long-term growth in life expectancy with concomitant improvements in the health and economic status of the population.

Third, the interactions among life expectancy, education and population dynamics are complex. The relationship between life expectancy and different dimensions of HC reveals a trade-off between investment in education and lifetime returns. Increases in life expectancy do not necessarily translate into improvements in HC per capita to the same extent. This contradiction is evident when increased schooling does not lead to increased economic participation or remaining years of work. Such dynamics suggest that increasing years of education may not always lead to commensurate gains in economic productivity or health outcomes.

Fourth, there are regional differences in the components of HC across high-income and G20 countries; these findings are consistent with the World Bank's HC index [70] and Lim et al [38]. In these regions, the components of HC—education, health and economic

participation—contribute synergistically to growth. Conversely, in other regions, sluggish progress in health and economic participation may hinder the effectiveness of investments in education, resulting in stunted HC growth. This variation underscores the need for integrated strategies that address all aspects of HC to promote equitable development.

## 5.  Conclusion

The inclusive wealth accounting of HC, as demonstrated through our analysis of 165 countries, significantly advances previous global HC assessments by integrating complex population dynamics and an NPV valuation method aligned with the capital budgeting paradigm. This approach enriches our understanding of the relationship between investment and returns in education and health and well aligns with policy needs, particularly in sustainable investment and impact assessment.

The results of this study highlight the crucial role of HC in health and economic participation—beyond education—in influencing HC accumulation. The data indicate that neglecting these aspects can lead to suboptimal HC development, especially in low-income regions. Consequently, there is an urgent need for a holistic policy approach that simultaneously addresses education, health, and economic factors to effectively harness the full potential of HC. Future strategies should account for the interdependencies between these factors to enhance HC and diminish regional development disparities.

Nevertheless, the insights garnered offer a sustainable perspective of HC development. While the correlation between HC growth and economic expansion is generally positive, our analysis identifies critical challenges in HC accumulation, particularly the need to focus on synergies between health and economic participation alongside education. Addressing these challenges is essential for fostering sustainable growth and equitably distributing development benefits.

The policy implications of the findings of this study are as follows. A positive relationship between HC and economic development is confirmed in this study on the basis of data derived from 165 countries, which suggests that improvement in HC is associated with economic development and health improvement. With respect to HC components, education has a favorable effect on economic development, and as the expectations regarding education increase within the population, economic development follows. The results suggest that policymakers should make sufficient financial investments in educational institutions to improve the education and skills of the population. However, the empirical evidence in this study shows that the effects of noneducational components of HC on health and economic outcomes present a time lag, indicating that comprehensive investment in HC is favorable for achieving sustainable economic development from a long-term perspective.

The limitations of this study are as follows. First, owing to data availability constraints, our life expectancy estimates are based on life tables rather than survival functions, and we use the Curtate method and assume that deaths are evenly distributed by age to estimate life expectancy alongside school life expectancy and working life expectancy by gender [61,71,72]. Compared with calculations via the complete method, life expectancy calculations via the Curtate method typically have an estimated error of half a year. However, the inclusion of integrated age-specific data on the schooling and labor force participation of the population ensures that our estimates reflect the consistency of education, health, and participation in the accounting system more accurately than previous accounts do, aligning well with the analytical requirements for this investigation (refer to Supporting Information S1 File). Furthermore, we set both the return rate to education and the discount rate at 0.085. The school life expectancy (SLE) and working life expectancy (WLE) are fixed to estimate the population's HC, which is sufficient for our cost–benefit analysis. However, dynamic estimations of

health, education, and economic participation status across all age groups could provide more detailed insights into each age cohort of the population, enhancing our financial and policy analysis. Future work should expand upon this by extending the data to conduct a comprehensive modeling exercise.

## Supporting information

**S1 File. Supporting_information.docx.** This file contains five figures (S1–S5) and three tables (S1–S3) related to the explanation of inclusive wealth framework, human capital dynamics, and related data sources.
(DOCX)

**S2 File. Data_upload.csv.** This file contains the detailed dataset of global human capital.
(CSV)

**S3 File. Wb_countries_admin0_10m.zip** . This file contains the World Bank Official Boundaries.
(ZIP)

## Author contributions

**Conceptualization:** shuning chen, Shunsuke Managi.

**Data curation:** shuning chen.

**Formal analysis:** shuning chen, Xiangdan Piao.

**Funding acquisition:** Shunsuke Managi.

**Investigation:** shuning chen, Xiangdan Piao.

**Methodology:** shuning chen.

**Project administration:** Shunsuke Managi.

**Resources:** shuning chen.

**Software:** shuning chen.

**Supervision:** Shunsuke Managi.

**Validation:** shuning chen, Xiangdan Piao.

**Visualization:** shuning chen, Xiangdan Piao, Jun Xie.

**Writing – original draft:** shuning chen, Xiangdan Piao, Jun Xie.

**Writing – review & editing:** shuning chen, Shunsuke Managi.

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
