## [Decision Letter · Decision Letter 0]

13 Aug 2024

PONE-D-24-23099Inclusive Human capital Accounting for 165 Countries Reveals the Critical Needs for Sustainability Investment.PLOS ONE

Dear Dr. Managi,

Thank you for submitting your manuscript to PLOS ONE. After careful consideration, we feel that it has merit but does not fully meet PLOS ONE’s publication criteria as it currently stands. Therefore, we invite you to submit a revised version of the manuscript that addresses the points raised during the review process.

We look forward to receiving your revised manuscript.

Kind regards,

Mingming Li

Academic Editor

PLOS ONE

Additional Editor Comments:

We note that one or more reviewers has recommended that you cite specific previously published works. As always, we recommend that you please review and evaluate the requested works to determine whether they are relevant and should be cited. It is not a requirement to cite these works. We appreciate your attention to this request.

Reviewers' comments:

Reviewer's Responses to Questions

**Comments to the Author**

1. Is the manuscript technically sound, and do the data support the conclusions?

Reviewer #1: Partly

Reviewer #2: Yes

2. Has the statistical analysis been performed appropriately and rigorously? 

Reviewer #1: No

Reviewer #2: Yes

3. Have the authors made all data underlying the findings in their manuscript fully available?

Reviewer #1: Yes

Reviewer #2: Yes

4. Is the manuscript presented in an intelligible fashion and written in standard English?

Reviewer #1: No

Reviewer #2: Yes

5. Review Comments to the Author

Reviewer #1: Title seems a sentences which should be revised.

Abstract

There is ample space to improve the language quality and writing pattern of the paper.

Introduction

This section is not according to standard academic writing. Unnecessary sub-headings and irrelevant discussion is there. Moreover, this section should be equipped with quoting the latest studies on the topic to convince the readers about the novelty and significance of the study.

There is absence of theoretical foundations for this study.

Absence of literature review makes the quality of the study, very poor.

Used methodology should be elaborated in more details.

The rationale to select the sample should be justified.

Each figure should be explained in details.

Discussion of the topic should be linked with earlier literature.

Theoretical, practical implications are also missing. Moreover, there is need to highlight the limitations and future prospects of the study.

Reviewer #2: Inclusive Human capital Accounting for 165 Countries Reveals the Critical Needs for Sustainability Investment

General comment

This study advances the Inclusive Wealth Human Capital Account to enhance global sustainable human capital assessment research. Authors have quantitatively measure human capital across 165 countries from 1990 to 2020 by incorporating education, health, and economic participation. Overall, I think the study is well-written, the topic is well-detailed, and I enjoyed reading it. I have some comments to improve the quality of the paper.

Major comments

1. Introduction Authors have detailed Introduction section. There are three sub-headings in the introduction section which are not necessary to show separately. I recommend you to remove the subsection headings by keeping the original texts. In addition, I suggest the author to read and cite Yamaguchi et al. (2019), Jingyu et al. (2020) and Islam and Managi (2021) literature for further improvement of the introduction section.

2. Method Some paragraphs are very short (for instance, the last paragraph of the method section) and my suggestion is to merge them with relevant paragraph.

3. Method Pay attention on writing the equations from 1 to 7. Why you need to put a dot (.) after every equations? To write a mathematical representations, sloppiness should be fundamentally removed.

4. Results At this stage, the result section is based on several plots of the panel data and their trend analysis. For robustness, I recommend the authors to consider econometric models to support your results with statistical significance.

5. Results Figure 6 should be improved in a way that X-axis and Y-axis information are clearly observed.

6. Discussion As suggested for the introduction section, the sub-section headings should be removed.

7. Conclusion The limitations of this study should be included in the final paragraph of the conclusion section.

8. Conclusion Expand the discussion on the policy implications of the findings in conclusion section.

Minor comments

1. I can observe many typos and grammatical inconsistency through the manuscript. I suggest the authors to remove it fundamentally.

2. Use the abbreviation for repeatedly used terminologies, for instance; HC can be used for Human Capital.

Overall evaluation

I think this paper made an effort to identify the necessity of incorporating complex population dynamics into human capital evaluations to better understand and enhance the inter-dependencies by aiming to reduce developmental gaps worldwide. I suggest the major revision before taking the final decision of this manuscript.

References

Islam, M. and Managi, S. (2021). Global human capital: View from inclusive wealth. In Measuring Human Capital, pages 39–54. Elsevier.

Jingyu, W., Yuping, B., Yihzong, W., Zhihui, L., Xiangzheng, D., Islam, M., and Managi, S. (2020). Measuring inclusive wealth of china: Advances in sustainable use of resources. Journal of Environmental Management, 264:110328.

Yamaguchi, R., Islam, M., and Managi, S. (2019). Inclusive wealth in the twenty-first century: a summary and further discussion of inclusive wealth report 2018. Letters in Spatial and Resource Sciences, 12:101–111.

6. PLOS authors have the option to publish the peer review history of their article (what does this mean? ). If published, this will include your full peer review and any attached files.

**Do you want your identity to be public for this peer review?** For information about this choice, including consent withdrawal, please see our Privacy Policy .

Reviewer #1: No

Reviewer #2: **Yes: ** Moinul Islam

---

## [Author Response · Author response to Decision Letter 1]

23 Oct 2024

To Reviewer #1

Thank you very much for your kind and beneficial comments, which have helped us in significantly improving the quality of our article. We sincerely appreciate it. We have carefully incorporated each comment to improve our manuscript.

Reviewer #1: Title seems a sentences which should be revised.

[Response] As suggested, we revised the title into “Inclusive Human capital Accounting for 165 Countries Reveals the Critical Needs for Sustainability Investment.”.

Abstract

There is ample space to improve the language quality and writing pattern of the paper.

[Response] We revised major parts of our manuscript and followed the article structure in the author guidance of PLOS ONE. Specifically, we restructured the introduction to explain the theoretical development of human capital assessment and clarify the research gap and contribution. We also revised methodology and the detailed description and discussion of the results as suggested in other comments.

We also did professional language editing to improve the quality of our manuscript.

Introduction

This section is not according to standard academic writing. Unnecessary sub-headings and irrelevant discussion is there. Moreover, this section should be equipped with quoting the latest studies on the topic to convince the readers about the novelty and significance of the study.

[Response] Thank you for your valuable feedback. We have thoroughly revised the introduction to align with standard academic writing conventions. Specifically, the following changes were made:

Removal of Unnecessary Sub-Headings and Irrelevant Discussions:

We have removed any unnecessary sub-headings and streamlined the content to ensure that the introduction flows smoothly without irrelevant discussions. The revised version focuses on providing a concise and coherent background, emphasizing the core themes of human capital (HC) and its relationship to economic growth and sustainable development.

Quoting the Latest Studies on the Topic:

To address your concern regarding the inclusion of more recent studies, we have incorporated several new references from recent empirical research, such as Liu et al. (2023), Angrist et al. (2021), Ziberi et al. (2022), Akcigit et al. (2024), and Zhang et al. (2023). These studies offer the most up-to-date insights into HC and its connection to economic growth and development. We believe this strengthens the introduction by highlighting the novelty and significance of the current research.

Focusing on the Novelty and Significance of the Study:

In the revised introduction, we have explicitly highlighted how our study builds on recent research and fills a gap in the literature by providing a comprehensive, global analysis that integrates health, education, and population dynamics into HC accounting. This addition clearly outlines the significance of our research, particularly in terms of how it updates the HC accounts of 166 countries and examines the regional variations and dynamics of HC in relation to sustainable development.

We believe that these revisions address your concerns and provide a stronger foundation for the study, aligning the introduction with academic standards and ensuring that the novelty and importance of the research are clearly communicated.

There is absence of theoretical foundations for this study.

[Response] Thank you for your insightful comment. We have strengthened the discussion of the theoretical foundations in both the introduction and methodology sections. Specifically:

Classical and Modern Human Capital Theory:

Our study is firmly grounded in well-established economic theories of human capital. The theoretical origins of HC can be traced back to Schultz (1961) and Becker (1964), who framed human capital as a key factor in economic growth. This was further developed through the endogenous growth models of Romer (1986) and Lucas (1988), which directly link human capital accumulation to long-term economic growth. These models emphasize the role of education, skills, and technology in driving economic prosperity, providing a strong theoretical basis for our analysis.

Jorgenson-Fraumeni Lifetime Income Approach:

In our methodology, we build on the well-known Jorgenson and Fraumeni (J-F) method for estimating human capital. This approach, widely recognized in economic literature, calculates human capital as the net present value (NPV) of future earnings streams. The use of the NPV model ensures that our study is consistent with wealth accounting frameworks, following the theoretical guidelines established by Dasgupta (2014, 2018) and Arrow et al. (2012).

Health and Population Dynamics in Human Capital Models:

We also incorporate recent theoretical advancements by recognizing health as a key component of human capital. This is rooted in Grossman's (1972) theory of health as a capital good that enhances productivity. Furthermore, we integrate population dynamics—including age structure, mortality rates, and migration patterns—based on the frameworks provided by Lee and Mason (2010) and Lutz and KC (2011). This ensures that our study not only considers traditional human capital components like education and skills but also accounts for the demographic and health-related factors influencing human capital.

By building on these well-established economic theories and methodologies, we believe our study has a strong theoretical foundation. We have revised the introduction and methodology sections to make these connections more explicit, and we hope this clarifies how the study is theoretically grounded.

Absence of literature review makes the quality of the study, very poor.

[Response] Thank you for your constructive feedback. Following the author guidance of PLOS ONE, we have now integrated a comprehensive review of the relevant literature directly into the introduction section. Specifically:

Incorporation of Recent and Relevant Studies:

The revised introduction includes references to a wide range of recent studies that address the relationship between human capital and economic growth (e.g., Liu et al., 2023; Angrist et al., 2021; Ziberi et al., 2022; Akcigit et al., 2024; Zhang et al., 2023). This review of contemporary research provides strong empirical support for the study’s focus on human capital's role in economic development.

Contextualizing Human Capital within Sustainable Development:

We have also integrated literature that explores the broader contributions of human capital to sustainable development. Key studies and global initiatives such as the World Bank’s Human Capital Index (HCI, 2018) and the United Nations Sustainable Development Goals (SDGs, 2015) are discussed to frame the importance of human capital in both economic and social contexts.

Gaps in the Literature and Contributions of the Study:

Additionally, we identify gaps in the existing literature, such as the need for a more comprehensive model that integrates health, education, and population dynamics in human capital accounting (e.g., Lim et al., 2018; Campbell and Üngör, 2020). By addressing these gaps, the revised literature review shows how our study contributes to the field by incorporating these factors into human capital estimations for 166 countries from 1990 to 2020.

We believe that these revisions have addressed the absence of a literature review and have significantly improved the quality of the study by grounding it in a thorough discussion of the most relevant and recent research.

References

Adawo, M. A. (2011). Has education (human capital) contributed to the economic growth of Nigeria?. Journal of economics and international finance, 3(1), 46.

Adelakun, O. J. (2011). Human capital development and economic growth in Nigeria. European journal of business and management, 3(9), 29-38.

Aghaei, M., Rezagholizadeh, M., & Bagheri, F. (2023). The effect of human capital on economic growth: The case of Iranâ s provinces. Quarterly Journal of Research and Planning in Higher Education, 19(1), 21-44.

Akcigit, U., Pearce, J., & Prato, M. (2024). Tapping into talent: Coupling education and innovation policies for economic growth. Review of Economic Studies, rdae047.

Arrow, K. J., Dasgupta, P., Goulder, L. H., Mumford, K. J., & Oleson, K. (2012). Sustainability and the measurement of wealth. Environment and development economics, 17(3), 317-353.

Bhargava, A., Jamison, D. T., Lau, L. J., & Murray, C. J. (2001). Modeling the effects of health on economic growth. Journal of health economics, 20(3), 423-440.

Bloom, D. E., Canning, D., & Sevilla, J. (2004). The effect of health on economic growth: a production function approach. World development, 32(1), 1-13.

Bouznit, M., Pablo-Romero, M. P., & Sánchez-Braza, A. (2023). Economic growth, human capital, and energy consumption in Algeria: evidence from cointegrating polynomial regression and a simultaneous equations model. Environmental Science and Pollution Research, 30(9), 23450-23466.

Cohen, D., & Soto, M. (2007). Growth and human capital: good data, good results. Journal of economic growth, 12, 51-76.

Dritsaki, M., & Dritsaki, C. (2024). The relationship between health expenditure, CO2 emissions, and economic growth in G7: Evidence from heterogeneous panel data. Journal of the Knowledge Economy, 15(1), 4886-4911.

Elkhalfi, O., Chaabita, R., Zahraoui, K., & El Alaoui, H. (2023). Public Spending on Human Capital and Economic Growth in Morocco. International Journal of Economics and Financial Issues, 13(4), 102-110.

Goldin, C. (2024). Human capital. In Handbook of cliometrics (pp. 353-383). Cham: Springer International Publishing.

Islam, M. and Managi, S. (2021). Global human capital: View from inclusive wealth. In Measuring Human Capital, pages 39–54. Elsevier.

Jingyu, W., Yuping, B., Yihzong, W., Zhihui, L., Xiangzheng, D., Islam, M., and Managi, S. (2020). Measuring inclusive wealth of china: Advances in sustainable use of resources. Journal of Environmental Management, 264:110328.

Liu, D., Wang, G., Sun, C., Majeed, M. T., & Andlib, Z. (2023). An analysis of the effects of human capital on green growth: effects and transmission channels. Environmental Science and Pollution Research, 30(4), 10149-10156.

Manca, F. (2012). Human capital composition and economic growth at the regional level. Regional Studies, 46(10), 1367-1388.

Mushkin, S. J. (1962). Health as an Investment. Journal of political economy, 70(5, Part 2), 129-157.

Siddiqui, A., & Rehman, A. U. (2017). The human capital and economic growth nexus: in East and South Asia. Applied Economics, 49(28), 2697-2710.

Yormirzoev, M. (2023). Human capital and economic growth in Central Asia. Post-communist Economies, 35(6), 533-545.

Zhang, Y., Kumar, S., Huang, X., & Yuan, Y. (2023). Human capital quality and the regional economic growth: Evidence from China. Journal of Asian Economics, 86, 101593.

Ziberi, B. F., Rexha, D., Ibraimi, X., & Avdiaj, B. (2022). Empirical analysis of the impact of education on economic growth. Economies, 10(4), 89.

Used methodology should be elaborated in more details.

[Response] Thank you for your valuable comment. In response, we have elaborated on the methodology to provide greater clarity and detail. Specifically, the following enhancements were made:

Detailed Description of Population Stages and Human Capital Calculation:

We have provided a more thorough explanation of how the population is grouped into three stages (0-4 years, school life expectancy, and post-education). The methodology now clearly explains how human capital is estimated at each stage, using the Mincer equation and the Jorgenson-Fraumeni (J-F) lifetime income approach.

Inclusion of Mathematical Formulations and Their Interpretation:

We have included detailed mathematical expressions for key components, such as the Net Present Value (NPV) formula for human capital, the estimation of school life expectancy (SLE), and work-life expectancy (WLE). Each equation is followed by a clearer explanation of its significance and the assumptions behind it.

Health and Population Dynamics Integration:

We have further clarified how health and population dynamics (e.g., age structure, survival rates, job participation) are integrated into human capital accounting. This includes the use of Markov models for calculating life expectancy and participation rates, as well as how health outcomes are factored into the overall framework.

Data Sources and Processing:

To improve transparency, we have elaborated on the data sources used in the study, including data from UNESCO, ILO, and United Nations Population Statistics, and the adjustments made to standardize school enrollment and labor force participation rates across countries.

These additions provide a more comprehensive understanding of the methodology used in the study, ensuring that it is fully transparent and replicable. We believe these revisions address your concern and significantly improve the clarity of the methodological framework.

The rationale to select the sample should be justified.

[Response] Thank you for your comment. We would like to clarify that our study is not based on a sample selection process but rather on a global human capital accounting framework. The study utilizes comprehensive data from 166 economies spanning 1990 to 2020 to provide a complete and comparative analysis of human capital trends across countries and regions.

We use multiple public global databases, including those from UNESCO, ILO, and United Nations Population Statistics, and supplement these with reliable national statistics or literature where necessary to ensure comprehensive global human capital accounting. While we acknowledge that there are some limitations in the fineness of the data, particularly in tracking changes in school enrollment and job market participation, the accounting framework we employ is flexible and allows for future improvements as more refined data becomes available. Moreover, the accounting results are consistent with those from other well-known studies, such as the World Bank's human capital accounting (2020) and Lee et al.(2018), further validating our approach.

By relying on this global accounting approach, we ensure that the study captures global trends and regional variations in human capital, making it more robust and generalizable than a sample-based analysis. We have revised the manuscript to clarify this methodological approach and believe it now better addresses your concern regarding the rationale for sample selection.

Each figure should be explained in details.

[Response] Thank you for your helpful feedback. In response, we have revised the manuscript to provide more detailed explanations of each figure, ensuring that the key insights and interpretations are clearly communicated. Specifically:

Detailed Explanation of Key Findings:

We have expanded the descriptions of each figure, focusing on the main findings and their relevance to the overall analysis. For example, in Figure 1, we now provide a clearer breakdown of the regional disparities in HC growth between 1990 and 2020, emphasizing the inverse relationship between initial HC levels and growth. In Figure 2, we further explain the correlation between HC and GDP per capita and the variations observed between 1990 and 2020 in different quartiles.

Clarifying Regional Trends:

For figures that compare regional trends (e.g., Figures 3, 4, 5), we have improved the explanations by highlighting how the different components of HC—such as education, adult labor proportion, and accounting price—vary across regions. For instance, in Figure 5, we explain the complex relationship between education and remaining working life across different regions, pointing out where inverse relationships are observed and how this affects overall HC accumulation.

Improved Discussion of Relationships Between Variables:

In response to your comment, we also explained the figures showing the variables' relationships. For example, Figure 4 now includes a more detailed discussion of how changes in school life expectancy (SLE), adult population share, and remaining working years contribute to overall HC growth and how these components are related to GDP growth.

Discussion of the topic should be lin

---

## [Decision Letter · Decision Letter 1]

7 Nov 2024

PONE-D-24-23099R1Human Capital and Regional Disparities: Advancing Accounting Frameworks with Education, Health, and Population DynamicsPLOS ONE

Dear Dr. Managi,

Thank you for submitting your manuscript to PLOS ONE. After careful consideration, we feel that it has merit but does not fully meet PLOS ONE’s publication criteria as it currently stands. Therefore, we invite you to submit a revised version of the manuscript that addresses the points raised during the review process.

We look forward to receiving your revised manuscript.

Kind regards,

Mingming Li

Academic Editor

PLOS ONE

Journal Requirements:

Additional Editor Comments:

Please polish your language as mentioned of reviewer before publication.

Reviewers' comments:

Reviewer's Responses to Questions

**Comments to the Author**

1. If the authors have adequately addressed your comments raised in a previous round of review and you feel that this manuscript is now acceptable for publication, you may indicate that here to bypass the “Comments to the Author” section, enter your conflict of interest statement in the “Confidential to Editor” section, and submit your "Accept" recommendation.

Reviewer #2: All comments have been addressed

Reviewer #3: All comments have been addressed

2. Is the manuscript technically sound, and do the data support the conclusions?

Reviewer #2: Yes

Reviewer #3: Yes

3. Has the statistical analysis been performed appropriately and rigorously? 

Reviewer #2: Yes

Reviewer #3: Yes

4. Have the authors made all data underlying the findings in their manuscript fully available?

Reviewer #2: Yes

Reviewer #3: Yes

5. Is the manuscript presented in an intelligible fashion and written in standard English?

Reviewer #2: Yes

Reviewer #3: Yes

6. Review Comments to the Author

Reviewer #2: Authors have taken account of the comments and modify the manuscript accordingly. I recommend the acceptance of the manuscript in PlosONE.

Reviewer #3: The manuscript should be proofread by an English native speaker. There are very long phrases in the text, that make the paper difficult to read. For example:

"Data required for cohort enrolment probabilities are provided by UNESCO Education Statistics (UIS) (2021),

which includes gross enrolment rates and enrolment rates for primary, secondary, and tertiary

schooling by gender." should be rephrased as: "Data required for cohort enrolment probabilities are provided by UNESCO Education Statistics (UIS) (2021). They include gross enrolment rates and enrolment rates for primary, secondary, and tertiary schooling by gender."

7. PLOS authors have the option to publish the peer review history of their article (what does this mean? ). If published, this will include your full peer review and any attached files.

**Do you want your identity to be public for this peer review?** For information about this choice, including consent withdrawal, please see our Privacy Policy .

Reviewer #2: No

Reviewer #3: No

---

## [Author Response · Author response to Decision Letter 2]

21 Nov 2024

To Reviewer #2

Reviewer #2: Authors have taken account of the comments and modify the manuscript accordingly. I recommend the acceptance of the manuscript in PlosONE.

We sincerely thank you for your positive evaluation of our manuscript and their recommendation for acceptance. We greatly appreciate your constructive feedback, which has helped us improve the manuscript.

To Reviewer #3

Reviewer #3: The manuscript should be proofread by an English native speaker. There are very long phrases in the text, that make the paper difficult to read. For example:

"Data required for cohort enrolment probabilities are provided by UNESCO Education Statistics (UIS) (2021), which includes gross enrolment rates and enrolment rates for primary, secondary, and tertiary schooling by gender." should be rephrased as: "Data required for cohort enrolment probabilities are provided by UNESCO Education Statistics (UIS) (2021). They include gross enrolment rates and enrolment rates for primary, secondary, and tertiary schooling by gender."

[Response] We sincerely thank you for your detailed comments and suggestions to improve the clarity of our manuscript. In response, we have:

(1) Carefully proofread the entire manuscript and sought feedback from a native English speaker to refine the language. (Please see the attached certification of proofreading.)

(2) Revised overly long sentences for better readability, including the specific example you highlighted. The revised text now reads:

"Data required for cohort enrolment probabilities are provided by UNESCO Education Statistics (UIS) (2021). They include gross enrolment rates and enrolment rates for primary, secondary, and tertiary schooling by gender."

We greatly appreciate your constructive feedback, which has significantly enhanced the quality and readability of our work.

---

## [Editor Report · Decision Letter 2]

22 Nov 2024

Human Capital and Regional Disparities: Advancing Accounting Frameworks with Education, Health, and Population Dynamics

PONE-D-24-23099R2

Dear Dr. Managi,

We’re pleased to inform you that your manuscript has been judged scientifically suitable for publication and will be formally accepted for publication once it meets all outstanding technical requirements.

Kind regards,

Mingming Li

Academic Editor

PLOS ONE
---

## [Editor Report · Acceptance letter]

PONE-D-24-23099R2

PLOS ONE

Dear Dr. Managi,

I'm pleased to inform you that your manuscript has been deemed suitable for publication in PLOS ONE. Congratulations! Your manuscript is now being handed over to our production team.

Kind regards,

on behalf of

Dr. Mingming Li

Academic Editor

PLOS ONE